# FINE-TUNING OFFLINE POLICIES WITH OPTIMISTIC ACTION SELECTION

## ABSTRACT

Offline reinforcement learning algorithms can train performant policies for hard tasks using previously-collected datasets. However, the quality of the offline dataset often limits the levels of performance possible. We consider the problem of improving offline policies through online fine-tuning. Offline RL requires a pessimistic training objective to mitigate distributional shift between the trained policy and the offline behavior policy, which will make the trained policy averse to picking novel actions. In contrast, online RL requires exploration, or optimism. Thus, fine-tuning online policies with the offline training objective is not ideal. Additionally, loosening the fine-tuning objective to allow for more exploration can potentially destroy the behaviors learned in the offline phase because of the sudden and significant change in the optimization objective. To mitigate this challenge, we propose a method to facilitate exploration during online fine-tuning that maintains the same training objective throughout both offline and online phases, while encouraging exploration. We accomplish this by changing the action-selection method to be more optimistic with respect to the Q-function. By choosing to take actions in the environment with higher expected Q-values, our method is able to explore and improve behaviors more efficiently, obtaining 56% more returns on average than the alternative approaches on several locomotion, navigation, and manipulation tasks.

## 1 INTRODUCTION

Offline reinforcement learning (RL) algorithms show strong performance on different tasks even when the available data contains sub-optimal behaviors such as random exploration or another RL agent's behavior. However, the quality of the offline data can limit the performance of an offline RL agent, and in these cases, further online fine-tuning can help improve the policy using additional environment interactions. A key challenge to online fine-tuning is that the training mechanisms designed for the offline setting are not effective at improving the policy online; offline learning requires a conservative objective to mitigate distributional shift between the learned policy and the behavior policy. However, during the online fine-tuning phase, the conservative objective causes the agent to keep doing the same behaviors, preventing it from exploring as needed to further improve the policy. In this paper, we aim to mitigate these challenges and develop a more effective method for offline-to-online RL.

This challenge of misaligned priorities between the offline and online training phases could in principle be tackled by switching training objectives when the online phase starts. For example, a policy trained with an offline RL algorithm such as CQL (Kumar et al., 2020) could be fine-tuned using an off-policy online algorithm designed for efficient exploration, such as SAC (Haarnoja et al., 2018). However, significantly changing the training objective between the offline and online phases can cause instability in the performance, potentially resulting in the destruction of behaviors learned during the offline phase (Nair et al., 2020). It can also exacerbate distributional shift between the offline data and the learned policy (Lee et al., 2022) since the more exploratory training objective will make the policy deviate from the offline behavior policy even more, resulting in unsatisfactory fine-tuning performance.

The problem of online fine-tuning from offline training presents an impasse: on the one hand, online fine-tuning requires efficient exploration, which is not feasible using the conservative offline training

objective; on the other hand, changing the objective might destroy the behaviors learned by the offline training. A key insight of our method is that we collect optimistic data without changing the training objective. To collect such exploratory data, we aim to use the knowledge embedded in the Q-function to direct exploration, i.e., selecting actions that are estimated to be better than the ones given by the policy. Concretely, during online fine-tuning, we can execute actions that have high estimated Q-values, thus providing exploratory data that can be used to improve the policy, while using the same offline objective with this new data.

The key contribution of this paper is an exploration technique that allows for stable and efficient offline-to-online fine-tuning. Our approach is simple, as it only modifies the action-selection process, and can in principle be implemented on top of any offline learning algorithm that doesn't explicitly penalize the Q-values of out-of-distribution state-action pairs. We show results built on top of LAPO (Chen et al., 2022) and IQL (Kostrikov et al., 2021) for several locomotion, navigation, and manipulation tasks, and find that by using this action-selection mechanism agents get an average of 56% more returns than the next best prior method.

## 2 RELATED WORK

**Exploration mechanisms:** One simple technique for exploration is to apply noise or randomness to the agent's behavior (Mnih et al., 2015; Lillicrap et al., 2015; Schulman et al., 2015). Other approaches explore in a more targeted way through novelty seeking (Houthooft et al., 2016; Bellemare et al., 2016; Pathak et al., 2017; Ostrovski et al., 2017; Fu et al., 2017a; Burda et al., 2018b;a) and entropy regularization (Pong et al., 2019; Eysenbach et al., 2018; Florensa et al., 2017; Gregor et al., 2016). These prior exploration methods are focused on online RL and are ill-suited for offline-to-online fine-tuning on their own, as exploration can negatively impact offline training. Our exploration mechanism is based on modifying the action-selection mechanism to choose actions more optimistically using a Q-function pre-trained with offline RL.

**Offline to online fine-tuning:** The problem of leveraging offline data during online RL has been widely studied especially when the offline data corresponds to demonstrations. Prior work has proposed a variety of approaches for this setting, ranging from online inverse RL and imitation learning (Lu et al., 2022; Ziebart et al., 2008; Finn et al., 2016; Fu et al., 2017b; Ho & Ermon, 2016; Kostrikov et al., 2018) to using the demonstrations to initialize the policy or replay buffer (Peters & Schaal, 2008; Vecerik et al., 2017; Hester et al., 2018; Rajeswaran et al., 2018; Zhu et al., 2018a; Gupta et al., 2019; Zhu et al., 2018b; Kober & Peters, 2009). We consider a more general case where the offline data may contain low or mixed quality data. The simplest approach for the general case of offline to online RL is to fine-tune online using the same offline objective (Kumar et al., 2020; Kostrikov et al., 2021; Nair et al., 2020; Lyu et al., 2022); however, this approach may be too conservative during the online phase, resulting in too little exploration and plateauing performance. Instead of maintaining the same offline training objective during the online phase, Wu et al. (2022) propose to gradually make the training objective less conservative. As we empirically show, this approach does not significantly improve exploration for the considered baselines. Lee et al. (2022) instead propose to target the over-conservatism of using an offline training objective during the online phase with a prioritized replay buffer that prefers more on-policy samples. This approach could in principle be combined with our method for enhanced exploration, but we leave this direction for future work. Unlike these past works, we develop a simple and sample-efficient fine-tuning approach that effectively trade-offs conservatism in the offline phase and optimism in the online phase without sacrificing stability.

**Offline meta reinforcement learning:** Offline meta RL is related to online fine-tuning, since it deals with efficient online adaptation using an offline dataset. However, the meta-RL domain assumes access to a *multi-task* offline dataset, and trains an agent to quickly adapt to new tasks (Mitchell et al., 2021; Dorfman et al., 2021; Pong et al., 2022). As opposed to offline meta RL, our work (and offline-to-online fine-tuning in general) does not assume access to multi-task data, and deals with adaptation and improvement in the context of the same offline task.

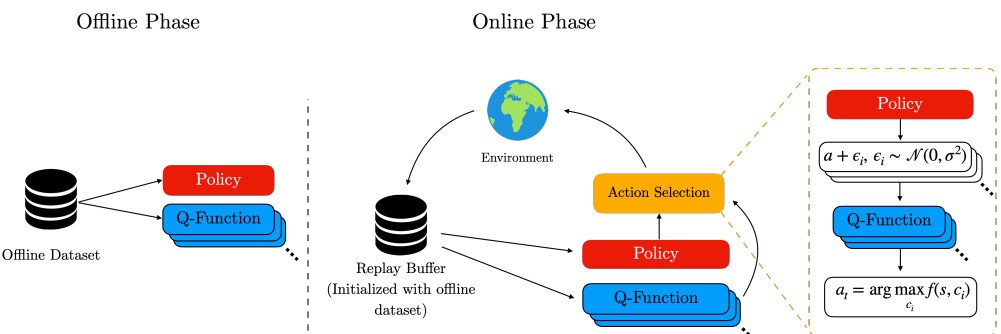

Figure 1: The online fine-tuning setting has two phases: an offline phase, in which no interaction with the environment is possible, and all learning has to be done using the offline dataset; and an online phase, in which the agent has to perform well from the beginning, and keep improving with access to the environment. Our method O3F trains a policy and a critic ensemble in the offline phase, and then in the online phase, fine-tunes the policy and critic ensemble while modifying how actions are selected. As illustrated on the right, actions are selected optimistically based on the estimated scores given by the scoring function from equation 2. For stability, the same training objective as the offline phase is used in the online phase, while incorporating the new online experiences into the replay buffer.

## 3    PRELIMINARIES

We address policy learning in a Markov decision process (MDP), defined by the tuple $(S, A, p, r)$, where $S$ and $A$ denote the state and action spaces. The unknown state transition probability distribution $p : S \times S \times A \to [0, \infty)$ represents the probability density of the next state $s' \in S$ given the current state $s \in S$ and action $a \in A$. The reward function assigns a sparse reward to every state-action transitions $r : S \times A \to \mathbb{R}$.

### 3.1    OFFLINE REINFORCEMENT LEARNING

In the offline RL setting, an agent only has access to a dataset of experiences $\mathcal{D} = \{(s, a, r, s')_i\}$ that are collected prior to the RL training using an unknown behavior policy $\pi_\beta$. Off-policy RL approaches can leverage the offline dataset to train a critic network (Q-function) $Q_\theta^\pi(s, a)$ with parameters $\theta$, which estimates the long-term discounted reward attained by executing action $a$ at state $s$ and following the policy $\pi$ thereafter. The critic network can then be used to update the policy in an actor-critic training scheme. The critic network can be trained using the following temporal difference learning objective:

$$\mathcal{L}(\theta) = \mathbb{E}_{(s,a,s') \sim \mathcal{D}}[(r(s, a) + \gamma \max_{a'} Q_{\hat{\theta}}(s', a') - Q_\theta(s, a))^2], \tag{1}$$

where $\hat{\theta}$ denotes the delayed parameters of the Q-function. Notice that $a'$ is a potentially out-of-distribution action, so $Q_\theta(s', a')$ might give an incorrect value. The $\max$ operator evaluated over many potentially-incorrect values is then likely to update the Q-function with incorrect information, resulting in sub-optimal policies. To alleviate this problem, prior offline RL methods propose two general solutions: (1) assigning low Q-values to the out-of-distribution actions by training a Q-function with a pessimistic objective (Kumar et al., 2020; Lyu et al., 2022), and (2) training the Q-function using equation 1 while constraining the policy to only produce actions contained in the offline dataset $\mathcal{D}$ (Kostrikov et al., 2021; Chen et al., 2022). The action-selection mechanism we present in this work relies on sampling the Q-function for slightly out-of-distribution state-action pairs and obtaining non-pessimistic value estimates. Our method is therefore only applicable to the second class of offline RL algorithms that constrain the actor, and not the critic.

### 3.2    OFFLINE-TO-ONLINE FINE-TUNING

As shown in Figure 1, having trained a policy and a Q-function with offline data, the goal of online fine-tuning is to further improve the behavior of the learned policy by allowing the agent to interact

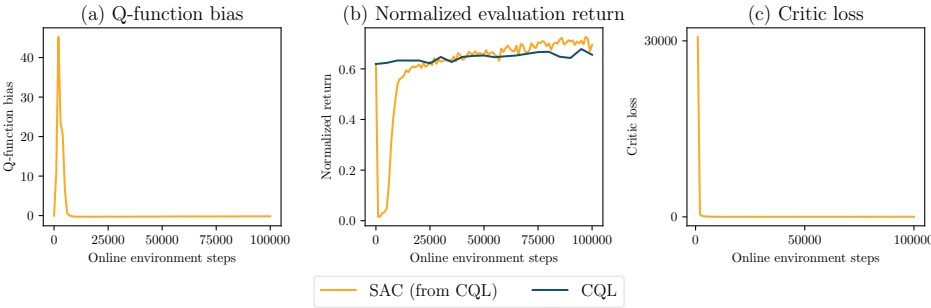

Figure 2: Average Q-function bias, performance, and critic loss for SAC initialized from CQL and trained with the halfcheetah-medium-v2 dataset in its replay buffer. The beginning of the online fine-tuning training destroys the performance CQL learned offline, and training with the optimistic SAC objective makes the Q-function greatly over-estimate state-action values.

with the environment. In every episode of the online fine-tuning phase, new samples are collected by unrolling the latest policy in the environment and are added to the replay buffer. Then, the policy and the Q-function are updated by the actor-critic learning algorithm which is employed for the online fine-tuning phase. Offline-to-online fine-tuning presents two closely related challenges: (1) conservative offline learning objectives limit agents' exploration power, which is vital for efficient online learning, and (2) changing training objectives when switching from the offline phase to the online phase can cause a significant drop in performance, due to distributional shift and the sudden change in objective. A suitable method for online fine-tuning should handle the transition from offline to online while improving the policy performance by carefully exploring the environment.

## 4 OBJECTIVE SWITCHING BETWEEN OFFLINE AND ONLINE PHASES

The pessimistic objectives in offline RL algorithms prevents proper exploration in online settings and limits the agent from efficiently improving its performance. As shown in Figure 2(b), online training with CQL does not notably improve the learning performance even though the agent is no longer bounded by the offline setting. This limited performance in the online setting can be explained by the restricted exploration capability of the agent. CQL avoids querying out-of-distribution actions, hence it stabilizes the offline training by explicitly assigning low Q-values to actions outside the support of the offline dataset. As a side-effect, CQL would have difficulties in efficiently exploring the environment, and therefore it demonstrates a poor performance at improving the policy in online settings.

As discussed earlier, a naive solution to this problem would be to pre-train a constrained policy with offline RL, and then modify the objective at the fine-tuning phase by lifting the constraints or making them less conservative. However, our empirical studies show that this approach do not perform well in practice. Figure 2(b) shows the results of using the soft actor-critic (SAC) algorithm for online fine-tuning of a policy trained with CQL. SAC leverages a maximum entropy objective to encourage exploration which is helpful to improve the performance at the online fine-tuning phase. However, as shown by the figure, fine-tuning the CQL policy with SAC leads to an immediate drop in performance, which takes about 15,000 online environment steps to recover. We hypothesize that this drop in performance stems from the sudden change in the Q-function objective (from CQL to SAC). As shown in figure 2(c), the critic loss spikes once the objectives changes. Additionally, as shown in figure 2(a), updating the objective results in 40x over-estimation of the Q-values for at least several thousands of steps after switching the objective. This renders the pre-trained Q-function sub-optimal and in need of more updates with online data. Updating the policy using this sub-optimal Q-function can destroy the behaviors that were learned during the offline training phase.

We may encounter a similar problem even for a minor modification of the objective. Figure 3 illustrates the online fine-tuning performance for CQL, AWAC, and IQL on the HalfCheetah environment with medium offline dataset for three settings: (1) no change in the objective, (2) changing the hyper-parameters at the beginning of the online fine-tuning phase to make the objective less

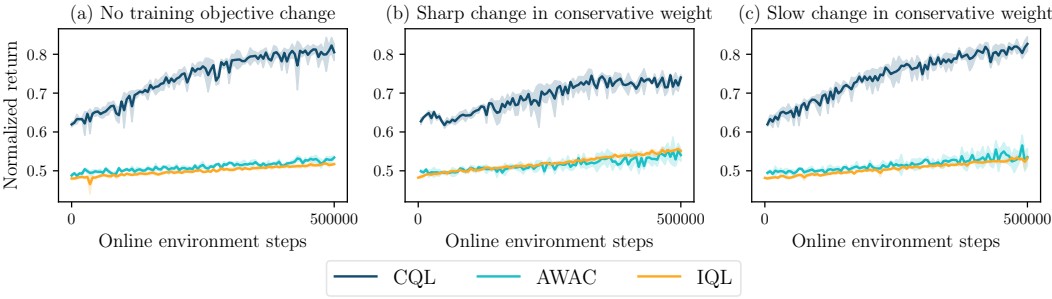

Figure 3: Comparison of results for online fine-tuning on halfcheetah-medium-v2 for CQL, AWAC, and IQL for 3 settings: (1) fine-tuning with the same offline training objective, (2) immediately changing the training objective to be less conservative at the online phase, and (3) linearly changing the hyper-parameters to make the training objective less conservative. For CQL we reduce the Lagrange threshold from 5 to 0.5, for AWAC we reduce the Lagrange multiplier from 1 to 0.1, and for IQL we increase the expectile from 0.7 to 0.85.

conservative, and (3) linearly updating the values of the hyper-parameters over time so as to make the change as smooth as possible. The hyper-parameters are the Lagrange threshold for CQL, the Lagrange multiplier for AWAC, and the expectile value for IQL. As shown by the figure, abruptly changing the CQL objective to be less conservative negatively affect the learning performance while smoothly updating the objective yields a similar level of performance as training with a fixed objective. IQL shows slightly better performance when the objective is updated abruptly, and AWAC training does not seem to be affected by the change in the hyper-parameter. To conclude, our empirical results show that making the objective less conservative during the online phase does not significantly improve the online fine-tuning performance.

## 5    OPTIMISTIC ACTION SELECTION MECHANISM

To explore efficiently without changing the offline training objective, we take advantage of the knowledge that the Q-function might contain about behaviors proximal to the ones the current policy produces. Provided that the Q-function has been trained so as to not explicitly penalize out-of-distribution actions (as, for instance, CQL would), we can expect the Q-function to sometimes assign reasonable values to *slightly out-of-distribution* state-action pairs. In this section we describe the offline to online optimistic fine-tuning (O3F) algorithm, an action-selection process that samples multiple action-candidates, and queries the Q-function to decide which action to explore.

### 5.1    OPTIMISM VIA ACTION RANKING

We leverage the knowledge embedded in the Q-function to choose actions that are better than the ones given by the policy. However, only slightly out-of-distribution samples from the Q-function can be trusted since value estimates for state-action pairs not supported by the training data can have arbitrarily-incorrect values. This motivates the design of our action-selection mechanism, as it only queries the Q-function for actions close to the ones given by the pre-trained policy. As demonstrated by Algorithm 1, O3F requires an offline dataset, a policy and an ensemble of critic networks pre-trained with the offline data, and the offline update rule for actor-critic training that is used for offline pre-training. We initialize a replay buffer with the offline dataset. In every step of online fine-tuning, O3F randomly samples $N$ candidate actions from a normal distribution, $c_i \sim \mathcal{N}(a, \sigma^2)$, where the mean $a$ is the action given by the policy, $a \sim \pi_\phi(s)$, and $\sigma$ is a small constant value. The candidate actions are then ranked according to some scoring function $f(s, a) \to \mathbb{R}$, and the candidate with the highest score gets executed in the environment. The new online experience is then incorporated into the replay buffer. After every step, O3F updates the actor and the critic using a batch of data from the replay buffer and the provided offline update rule.

## 5.2 UNCERTAINTY OPTIMISM SCORING FUNCTION

We consider two design choices for the scoring function $f(s,a)$: (1) the estimated Q-value, i.e., $f(s,a) = \frac{1}{m}\sum_{j=1}^{m}\left[Q_{\theta_j}(s,a)\right]$, and (2) an upper confidence bound of the estimated Q-values which we refer to as the *uncertainty optimism* scoring function. In both cases, O3F only queries actions with trusted values, i.e., actions close to the policy output. This is particularly important for the uncertainty optimism scoring function, as the uncertainty of the value estimates farther away from the policy output can be arbitrarily high, which results in favoring out-of-distribution actions.

---

**Algorithm 1** O3F

---

**Require:** Offline algorithm Update$(\phi^{(t)}, \{\theta_i\}^{(t)}, \mathcal{D})$
**Require:** Offline policy $\pi_\phi$, Critic ensemble $\{Q_{\theta_i}(s,a)\}$
**Require:** Offline dataset $\mathcal{D}$
1: **for** step $t$ in $1, \ldots, T$ **do**
2:      $a \sim \pi_\phi(a|s_t)$
3:      Sample $n$ action-candidates $c_i \sim \mathcal{N}(a, \sigma^2)$
4:      Get best candidate ranking by expected Q-value.
         $a_t = \arg\max_i \frac{1}{m}\sum_{j=1}^{m}\left[Q_{\theta_j}(s_t, c_i)\right]$
5:      $s_{t+1} \sim p(s_{t+1}|s_t, a_t)$
6:      $\mathcal{D} \leftarrow \mathcal{D} \cup \left\{(s_t, a_t, r(s_t, a_t), s_{t+1})\right\}$
7:      $\phi, \{\theta_i\} \leftarrow \text{Update}(\phi^{(t)}, \{\theta_i\}^{(t)}, \mathcal{D})$
8: **end for**

---

We model the uncertainty of the value-estimates by using an ensemble of independently-trained Q-functions. Let $\{Q_{\theta_i}(s,a) \rightarrow \mathbb{R}\}$ be the ensemble of Q-functions with parameters $\{\theta_i\}$. We define $Q_{\text{mean}}(s,a) = \frac{1}{m}\sum_{i=1}^{m}\left[Q_{\theta_i}(s,a)\right]$, and $Q_{\text{std}}(s,a) = \sqrt{\frac{1}{m}\sum_{i=1}^{m}|Q_{\theta_i}(s,a) - Q_{\text{mean}}(s,a)|^2}$. The proposed scoring function is then the upper confidence bound given by (Chen et al., 2017; Ghadirzadeh et al., 2016):

$$f(s,a) = Q_{\text{mean}}(s,a) + \omega Q_{\text{std}}(s,a), \tag{2}$$

where $\omega$ is a hyper-parameter balancing the optimistic level of the scoring function. For two action candidates with similar estimated values, the upper confidence bound selects the action with higher prospects to encourage exploration.

### 5.3 IMPLEMENTATION DETAILS FOR O3F

We evaluate our method O3F applied on top of two offline RL algorithms, LAPO (Chen et al., 2022) and IQL (Kostrikov et al., 2021). For both algorithms, the offline training phases are carried out in a similar fashion to the original works. For LAPO, we find that training the critic in a manner similar to Chen et al. (2021) improves sample-efficiency. In particular, we train an ensemble of 5 Q-functions, and use a random subset of the Q-functions to generate the training target. This modification lets us update the critic 20 times per environment step during the online fine-tuning phase. This style of critic-training made IQL unstable, and thus we do not include it in the final method for IQL. Instead, IQL's critic is trained the same way as the original paper, with the distinction that we train 5 Q-functions instead of 2 to be consistent with the LAPO-based implementation.

Our method introduces 3 hyper-parameters: $N$, the number of action-candidates to generate; $\sigma$, the standard deviation of the normal distribution; and $\omega$, the optimism weight for scoring function. We found that $N = 100$ works well on every considered task. On locomotion tasks, we found $\sigma = 0.2$ to yield the best performance, and every other task uses $\sigma = 0.05$. We hypothesize that because of the dense reward function of the locomotion tasks, the Q-function can potentially be estimated more globally which enables O3F to evaluate a wider range of actions, and not only the ones located close to the policy output. On section 6.3 we compare the performances of $\omega = 1$ and $\omega = 0$, and find that the uncertainty optimism scoring function improves performance for the halfcheetah-medium-v2 task, decreases performance on kitchen-mixed-v0, and maintains performance on antmaze-large-diverse-v1. To avoid unnecessary complexity, the rest of the experiments use $\omega = 0$.

## 6 EXPERIMENTAL EVALUATION

The goal of our experiments is to, first and foremost, evaluate how O3F compares to prior methods. Moreover, we aim to study whether O3F is compatible with multiple base offline RL methods and to ablate the role of different design choices within O3F.

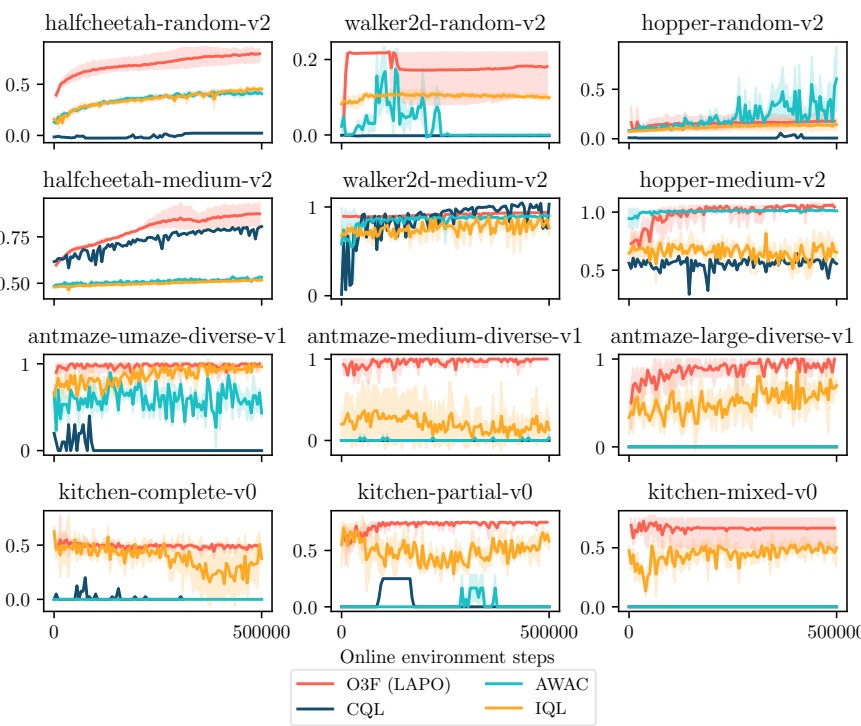

Figure 4: Normalized scores on D4RL tasks. Our method implemented on top of LAPO outperforms prior methods on the challenging Antmaze and Kitchen tasks, and either outperforms or performs comparatively to prior methods on locomotion tasks with random and medium datasets.

## 6.1 EXPERIMENTAL SET-UP

**Tasks.** We evaluate O3F on a variety of locomotion, manipulation, and navigation tasks from the D4RL benchmark Fu et al. (2020). We consider twelve total problems: six in simple environments and six in more complex environments. For the simple tasks, we use the D4RL Gym-MuJoCo locomotion tasks consisting of the HalfCheetah, Hopper, and Walker2D domains. Since we want to evaluate the ability to improve with online experience, we focus on the random and medium offline datasets, which have the lowest offline performance (as opposed to, e.g., using the expert dataset, which leaves little room for further improvement). The random dataset consists of experiences collected by a random policy, while the medium dataset uses experiences collected by a mediocre policy trained with early stopping. Since the locomotion tasks are relatively simple and straightforward, we also evaluate on the ant-maze and Franka kitchen environments, which are considerably more difficult. The ant-maze domain requires both learning to walk, as well as reaching the goal in a maze. We consider the three diverse ant-maze datasets, which involve mazes of increasing difficulty. The Franka kitchen domain involves controlling a simulated Franka robot in a kitchen environment solving a sequence of tasks. We consider all three Franka kitchen datasets in the D4RL benchmark: "complete", "mixed", and "partial".

**Comparisons.** We evaluate the performance of O3F implemented on top of both LAPO (Chen et al., 2022) and IQL (Kostrikov et al., 2021) in comparison to the prior work on offline-to-online fine-tuning. In particular, we consider three strong prior methods as baselines: (1) CQL (Kumar et al., 2020), (2) IQL (Kostrikov et al., 2021) and AWAC (Nair et al., 2020). CQL and IQL are used in offline-to-online fine-tuning settings by adding new experiences to the replay buffer and training with the same offline procedure provided all offline and online data. AWAC (Nair et al., 2020) is specifically designed for the online fine-tuning setting.

For CQL and AWAC, we use d3rlpy (Takuma Seno, 2021), an offline RL library that provides implementations of multiple offline RL algorithms, and allows for online fine-tuning. For IQL, we use the

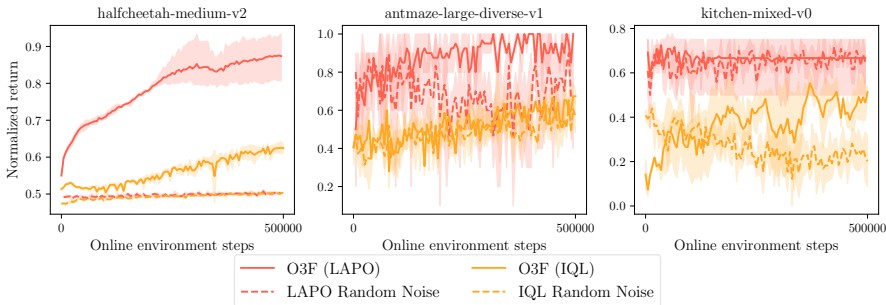

Figure 5: Comparison of O3F applied on top of both LAPO and IQL to a baseline of the base algorithm with action noise during online data collection. We find that O3F is compatible with both LAPO and IQL, and outperforms LAPO and IQL with random noise in two out of three domains while matching the performance in the third domain.

original paper's implementation. For all three prior methods, we use the original hyper-parameters, including the policy and the critic function network architecture. Every experiment starts with training a model only using the offline dataset, like the original work. Then, we initialize a replay buffer with the offline dataset, and proceed to alternate between taking a step in the environment and adding the experience to the replay buffer, and training the agent with the original offline procedure.

## 6.2 Main Results

Figure 4 shows the comparison of our method (O3F on top of LAPO) to the three prior methods. Comparisons of O3F with LAPO and IQL are made in section 6.3. O3F performs comparably to or better than all of the prior methods in all the tasks except for hopper-random-v2, in which AWAC obtains the best performance. O3F generally does not see a drop in performance after the start of the online fine-tuning phase, which shows that acting optimistically after the offline training phase can improve performance without sacrificing stability. We find that CQL and AWAC are unable to solve the Franka kitchen and AntMaze tasks, with the exception of AWAC on the simplest AntMaze setting (umaze), while O3F on top of LAPO reliably outperforms IQL on these tasks. For the simpler locomotion tasks, O3F outperforms all the other methods for halfcheetah-random-v2, walker2d-random-v2, and halfcheetah-medium-v2, performs comparatively to the best prior method on walker2d-medium-v2 and hopper-medium-v2, and underperforms AWAC for hopper-random-v2. On average, O3F performs 56% better than the next best method (IQL).

## 6.3 Further Comparisons and Ablations

**Action candidate ranking**: We compare O3F with the simple baseline of adding random noise to the actions. O3F on top of both LAPO and IQL implements the action-selection mechanism described in Section 5.1 and the implementation details in Section 5.3. In contrast, we compare to the original approaches of LAPO and IQL with noise added to the actions during online data collection, i.e. with noise $\epsilon \sim \mathcal{N}(0, \sigma^2)$ (where $\sigma$ is the same as in O3F). Results for this experiment can be seen in Figure 5. First, we find that O3F is compatible with both LAPO and IQL, matching or outperforming the performance of the base algorithm. For LAPO, our approach improves performance on halfcheetah-medium-v2 by 58%, on antmaze-large-diverse-v1 by 28%, and maintains the performance for kitchen-mixed-v0, where performance was already close to saturation (we find that LAPO does not surpass 0.75 returns on kitchen-mixed-v0 even with further fine-tuning.) For IQL, O3F improves performance by 14% on halfcheetah-medium-v2, and by 39% on kitchen-mixed-v0, while maintaining performance on antmaze-large-diverse-v1.

**Uncertainty optimism scoring function**: We compare two designs for the scoring function in our method: (1) scoring action candidates by estimating their value using the Q-function ensemble (this corresponds to $f_\phi(s, a)$ from Eq. 2 when $\omega = 0$); and (2) the uncertainty optimism scoring function introduced in section 5.2 ($f_\phi(s, a)$ when $\omega = 1$). We show results for 3 environments (halfcheetah-medium-v2, antmaze-large-diverse-v1, and kitchen-mixed-v0) on Figure 6. The uncertainty opti-

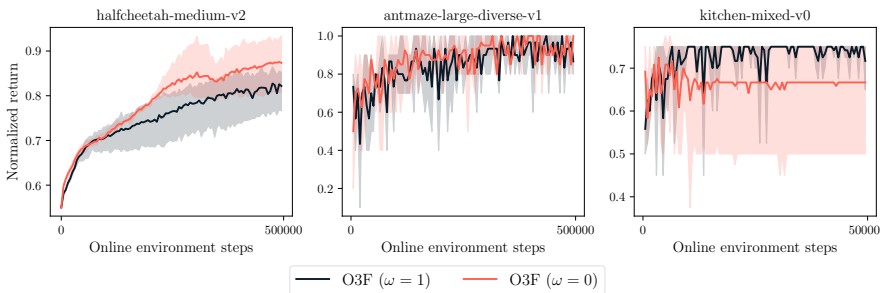

Figure 6: Evaluation of two different scoring functions for action candidates. $\omega = 1$ corresponds to the uncertainty optimism scoring function, which prefers actions with higher q-value uncertainty. $\omega = 0$ doesn't take uncertainty into account. We find that the uncertainty optimism scoring function performs best in the Franka kitchen domain, but is less advantageous in the other two domains.

mism scoring function outperformed the baseline scoring function on the kitchen environment, but it underperformed in the halfcheetah-medium-v2 environment. On the antmaze-large-diverse-v1 there weren't significant differences in performance between the two scoring functions. To avoid unnecessary complexity, we focus our experiments on the baseline scoring function ($f_\phi(s,a)$ where $\omega = 0$).

## 7 CONCLUSION

We presented a simple yet efficient offline-to-online fine-tuning strategy to further improve a pre-trained offline policy. The goal of online fine-tuning is to improve the agent's performance by properly exploring the environment without forgetting the behaviors already learned by the offline training. However, most offline RL algorithms are pessimistic in that they prohibit the agent from exploring state-action pairs not supported by the offline data. Therefore, it is challenging to fine-tune the learned policy to acquire new behaviors. We empirically showed that fine-tuning with the same objective does not result in a satisfactory fine-tuning performance, and making the objective less conservative negatively affects the behaviors already learned by the offline training. We presented O3F, an offline to online optimistic fine-tuning algorithm which facilitates exploration without changing the objective. O3F improves exploration by using an action-selection method which favors optimistic actions with respect to a Q-function pre-trained with offline data. We empirically showed that fine-tuning with O3F can on average achieve 56% more returns compared to IQL, AWAC and CQL baselines on several locomotion, navigation and manipulation tasks.

**Limitations:** There are two important limitations for the proposed algorithm: (1) it is only applicable to offline RL methods that do not explicitly penalize the Q-values of out-of-distribution state-action pairs, and (2) it can only refine the behaviors already learned in the offline phase, without the capability of discovering new behaviors. We argue that the second limitation is generally applicable to all fine-tuning methods that are built on top of pessimistic offline RL algorithms. We will study this limitation in more details as a part of our future work.

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
