# OpenReview forum: "Fine-Tuning Offline Policies With Optimistic Action Selection"
_ICLR.cc/2023/Conference — Submitted to ICLR 2023_

### Official Review · Reviewer_ggWX · 2022-10-23

**Confidence:** 4
**Correctness:** 2
**Technical Novelty And Significance:** 3
**Empirical Novelty And Significance:** 3
**Recommendation:** 3

**Clarity, Quality, Novelty And Reproducibility:**

The clarity of the presentation is good.

The experimental quality has several issues as discussed above.

The proposed algorithm is novel to my understanding.

The code is provided to reproduce the results, but I have not tried to do so myself. It also seems from looking at the code that the baselines are not included and neither is any information about hyperparameter selection.

**Strength And Weaknesses:**


### Strengths

1. The proposed algorithm is simple and elegant. Moreover, the motivation is reasonable, saying that we can get better performance by encouraging better exploration without forgetting the information learned during offline training.

2. Experiments are presented on a variety of tasks from the D4RL suite, that seem to show consistent gains.

### Weaknesses

1. The most serious issue with the paper is the improper choice of baselines. First and foremost it is especially strange that the main experiment does not include LAPO as a baseline. Since the main claim of the paper is that the O3F action selection mechanism is the important innovation, this should be isolated. This means that either the main result should use LAPO as a baseline, or examine O3F on top of IQL. Second, neither of the most closely related works, Wu et al 2022 and Lee et al. 2022 that are cited in the introduction, are used as baselines in the experiments (excluding the comparison in Figure 3 on one task with a different base learner than the one used later on).

2. Some relevant ablations are missing. Figure 5 is a good start on an ablation showing that the action selection is better than adding random noise to the actions. The other relevant comparisons that should be added are to (a) sample actions directly from the distribution output by the learned policy, and (b) choose deterministic actions from the mean of the output of the learned policy. These ablations would give a more fair comparison to what happens without the exploration that O3F claims to provide.

3. There are no details about hyperparameter tuning and it seems that there was possibly substantially more hyperparameter tuning for O3F than for the baselines. Specifically, from the paper there seem to be several hyperparamters that are potentially tuned for the method differently than baselines including the size of the Q ensemble, number of Q updates, number of randomly sampled actions, and the variance of the gaussian noise.

4. There are no experiments verifying that the improved performance is indeed due to improved exploration as is frequently claimed in the paper. Such an experiment by e.g. measuring the amount of coverage during online training with and without the O3F action selection mechanism would help to better connect the motivation with the results.


**Summary Of The Paper:**

This paper proposes offline to online optimistic finetuning (O3F), an algorithm for offline to online RL. The algorithm maintains the same offline learning objective during online learning, but uses a different action distribution that selects actions with potentially higher Q values. This is achieved by simple adding gaussian noise to the mean action output by the policy network and then selecting the action with highest Q value. Experiments are run on a variety of datasets from the D4RL suite, showing better performance than some baselines.

**Summary Of The Review:**

Overall I think this paper addresses an interesting problem and proposes a potentially clever algorithm. However, as implemented, I do not have confidence in the quality of the experimental results and rate the paper as a reject. If the authors provide improved baselines and comparison of hyperparameter tuning, I would consider raising my score.

---

> ### Author Response · Authors · 2022-11-19
> **Response to Reviewer ggWX**
>
> Thank you for providing valuable feedback on the paper. We will use it to revise the paper. We address the raised issues below:
>
> - Re improper choice of baselines, and missing relevant ablations: we will include a common response to all reviewers addressing these issues shortly.
>
> - For the methods that use RedQ updates, the Q-ensemble size is always fixed at 5, and the number of Q-updates is always 20 (we did not perform hyperparameter search for these hyperparameters). For the number of randomly sampled actions and variance of Gaussian noise we hope that the ablations in the next version of the paper will address these concerns.
>
> - Re "There are no experiments verifying that the improved performance is indeed due to improved exploration": Thank you for the valuable suggestion.

---

### Official Review · Reviewer_NH1h · 2022-10-25

**Confidence:** 4
**Correctness:** 3
**Technical Novelty And Significance:** 1
**Empirical Novelty And Significance:** 2
**Recommendation:** 3

**Clarity, Quality, Novelty And Reproducibility:**

The paper is clearly written, and easy to follow. However, I would say the technical/presentation quality is not at a good level since the arguments and the algorithms are not well aligned. The problem setting is quite original and interesting, but the proposed algorithm is not very original.

**Strength And Weaknesses:**

Strength:
- interesting problem setting
- a simple practical method that works

Weakness:
- interesting motivation but proposed algorithm is not aligned to it:

This paper starts with the motivation that the critics and the policies learned by pessimistic offline RL algoritms will not in general be improved well in the online fine-tuning case. The main example the paper presents is CQL, which has an explicit regularization term that makes Q conservative outside the data region, having lower value than the true Q value. It would have been the source of Q-function bias in Figure 2-(a), because suddenly removing the regularization will make all the Q estimates outside the data support invalid. In Figure 3 the paper shows all CQL, AWAC, IQL struggles when applied to online to offline fine-tuning and the objective suddenly changes to become non-pessimistic.

However, the proposed algorithm O3F does not deal with such critic objective change, and all the interesting motivation suggested above is actually not addressed in the algorithm. O3F also mainly deals with LAPO algorithm, which does not seem to have an explicit regularization that gives a pessimism. Does it even have a similar problem that CQL suffers as mentioned above? I assume not.

- Lack of originality of O3F:

The proposed algorithm O3F is now simply an action selection by first sampling actions from policy and ranking them according to upper confidence bounds. Selecting actions by ranking them according to Q values is a well known practical method used in BCQ/BEAR. Selecting actions by upper confidene bound is also a well known method. The suggested O3F is combination of those two, and no theoretical properties of such algorithm is desribed in the paper.

- Experiment that does not clearly show the improvement of O3F:

Since O3F is simply an action selection mechanism in online fine-tuning phase, I think the authors should have given more focus on a same algorithm with different action selections. In the result of figure 4, LAPO seems to simply outperform all the other algorithms from the start, and it is hard to see the improvement made by O3F. In figure 5 and 6, the difference between action selection algorithms are shown, but it is only shown on 3 domains. Although I agree on that the proposed algorithm will lead to performance improvement, the experiments do not support that well.


**Summary Of The Paper:**

This paper proposes O3F, which is an action selection algorithm when doing an offline-to-online fine tuning. Since value estimates and policies we had was trained under a conservative objective (in many offline Rl algorithms), it may be too pessimistic in online fine-tuning, and may not improve much. To alleviate this issue, this paper proposes to select actions according to the upper confidence bound of Q functions.

**Summary Of The Review:**

Overall, the problem setting and the motivation part were interesting, but the proposed algorithm O3F is not an algorithm that properly addresses the issues presented. The experiments are not also enough to show the improvement of O3F. In these sense, I believe that the contribution of this paper is limited and lean toward rejection.

---

> ### Author Response · Authors · 2022-11-19
> **Response to Reviewer NH1h**
>
> Thank you for providing valuable feedback on the paper. We will use it to revise the paper. We address the raised issues below:
>
> - Re "interesting motivation but proposed algorithm is not aligned to it": In the paper we argue that changing the critic objective can lead to training instability or insufficient exploration. O3F deals with this issue by not changing the critic objective at all, and instead uses an optimistic action-selection mechanism as a source of exploration.
>
> - Re "Experiment that does not clearly show the improvement of O3F": we will include a common response to all reviewers addressing these issues shortly.

---

### Official Review · Reviewer_jHot · 2022-10-25

**Confidence:** 3
**Clarity, Quality, Novelty And Reproducibility:** See above.
**Correctness:** 3
**Technical Novelty And Significance:** 3
**Empirical Novelty And Significance:** 2
**Recommendation:** 5

**Strength And Weaknesses:**

Strength:

1. The topic is interesting and potentially useful in practical scenarios.

2. The proposed method is simple but seems to be new, and it makes intuitive sense. Specifically, the exploration angle to accelerate online RL seems to be new.

3. The presentation is reasonably clear.

Weaknesses:

The paper is mostly empirical, which is fine. But I want to see more empirical verifications, which I consider critical to make the proposed method persuasive.

The central goal is to use offline data to accelerate online RL, and the acceleration is done from the exploration perspective.

As a result, I think it is important to show a baseline without using offline data at all but using an ensemble for exploration. This is critical to validate the utility of offline data.

To verify such an ensemble approach is needed for exploration, it is also important to have a baseline using a naive & computationally more efficient exploration (say, do CEM only without using the score function computed by the ensemble) method.

I am willing to adjust my score based on the author’s response and other reviews.

**Summary Of The Paper:**

The paper focuses on how to use offline data to accelerate online learning. The authors consider this problem from an exploration perspective. The intuition is that offline RL algorithms typically estimate action values conservatively to avoid overestimation errors due to OOD actions. Such conservative estimates potentially hurt exploration when transferring to an online learning setting. As a result, the authors propose to estimate the variance of action value estimates (which is done by using an ensemble of action value networks). The agent can take actions according to Q + variance term, similar to a UCB-style algorithm. The paper emphasizes that simple modifications of offline RL algorithms cannot do well when transferring to an online RL setting. Hence there is a strong motivation to use the proposed method. Experiments on frequently used offline RL testing beds are conducted to verify their effectiveness.


**Summary Of The Review:**

The paper studies an interesting topic and proposes an intuitive and simple idea to achieve its goal. It lacks some important empirical results. I am willing to adjust the score based on the author's response.

---

> ### Author Response · Authors · 2022-11-19
> **Response to Reviewer jHot**
>
> Thank you for providing valuable feedback on the paper. We will use it to revise the paper. We will include a common response to all reviewers shortly which will address the issues brought up, including more empirical verifications and choice of baselines.

---

### Official Review · Reviewer_b1rp · 2022-11-03

**Confidence:** 4
**Correctness:** 2
**Technical Novelty And Significance:** 2
**Empirical Novelty And Significance:** 2
**Recommendation:** 3

**Clarity, Quality, Novelty And Reproducibility:**

Clarity: The work clearly states the problem and the proposed method. The main concerns are in the experimental part, and the results do not clearly provide evidence to support the proposed method.

Quality: The work is well-written and didactic. The experimental part could be greatly improved, as pointed out by the weaknesses above. It misses the discussion of some important limitations, although it brings important ones in the Conclusion section. One critical concern is: given such limitations, how applicable/effective is the proposed method for a broader set of problems?

Novelty: The method is simple and its novelty relies on integrating known pieces (gaussian action selection, critic ensemble, disagreement metrics) in the context of offline-to-online finetuning.

Reproducibility: The work provided the source code to encourage reproducibility. In the experimental section, it is not clear to me the statistical significance of the results in Figure 2. Additionally, there is not any mention of the computational cost. This is important because O3F samples 100 actions to be evaluated by 5 critic networks in each environment iteration. This seems to increase the computational cost considerably.


**Strength And Weaknesses:**

Strengths:

- The online fine-tuning of offline pre-trained policies is a very important topic for RL nowadays, which makes the proposed work well-motivated;
- The proposed solution is simple and seems to be effective. This makes it suitable for practitioners and real-world problems that rely on the same assumptions;

Major Concerns:
- The proposed method relies on strong assumptions that are not discussed in the paper. First, the gaussian parametrization relies on a continuous action space. Secondly - and most important - it relies on a continuity property of the Q-functions (trained on the behavior policy data) for “closer” actions (i.e., numerically similar actions would have similar outcomes for the same state). This should be reasonable for some environments, but it is hard to evaluate the generality of this assumption without any evidence.

- In the first paragraph, the work states as motivation the fact that fine-tuning with conservative objectives will cause the agent to keep doing the same behaviors, hurting the exploration needed for policy improvement. Nevertheless, one of the limitations of the proposed method (stated in Section 7) is that it only refines already learned behaviors and does not discover new ones. From this perspective, O3F does not seem to satisfy one of its main motivations. It is important to clearly state the challenges mitigated by the proposed method.

- Figure 2 aims to provide empirical evidence for the proposed problem. However, there are some questionable methodological points. First, the results do not seem to show results with confidence intervals among different seeds, so it is not possible to assess statistical significance. Secondly, it is not clear why the baseline method is CQL, especially given that it is not even in the class of offline RL methods leveraged by O3F. Why not consider LAPO/IQL? Finally, it would also be helpful to show the results from standard, online SAC from scratch, as a sanity check to validate that the offline training is indeed necessary for this task.

- Figure 3 aims to empirically refute the hypothesis that making the objective less conservative improves online fine-tuning. In this case, it is crucial to report the results from Wu et. al. [1], since it provides empirical evidence in the opposite direction. Perhaps the proposed heuristics in Figure 3 are not good enough to prove this hypothesis. A strong argument needs to be grounded on state-of-the-art results.

- From Figure 4, it is difficult to credit the final good performance to the online finetuning or to the offline training, because the offline training methods differ. This causes different starting performances during online fine-tuning. If the idea here is to just compare the final best performance, it would be better to have a table with asymptotic performance. Furthermore, it is necessary to consider an online method (SAC) as a sanity baseline, to make sure the offline pretraining is indeed necessary for such environments. Also, you should also consider LAPO-only and O3F on top of IQL, to better understand the impact of the proposed method.

- Figure 5 brings the most critical result, as it ablates the proposed method. Therefore, it is necessary to show the results in all environments (not only 3) to give enough empirical support. It also needs to show the performance of LAPO-only and IQL-only, which are the offline-only baselines.

- Figure 6: analogous to the previous figure, it would be interesting to provide results from all environments. The current report shows mixed results, and it is not clear if the uncertainty is indeed necessary. Given that most of the reported results do not account for the uncertainty optimism scoring function, the argument towards “optimistic” action selection is weak.

Further Suggestions/Minor Concerns:

- There is no study on the gaussian noise variance. It would be interesting to have any notion about how sensible is this parameter, especially because online fine-tuning often requires sample efficiency and hyperparameter tuning is prohibitively expensive.


References

[1] Wu el. al. Supported Policy Optimization for Offline Reinforcement Learning. NeurIPS, 2022.


**Summary Of The Paper:**

This work studies the problem of finetuning policies that were previously trained on offline datasets, a challenging problem since offline pre-training often leverages pessimistic objectives, which would hurt online fine-tuning in searching for new behaviors. To approach this, it proposes O3F, a method that works on top of some classes of offline RL algorithms by sampling actions based on a gaussian distribution and ranks them according to a critic ensemble and their disagreement. The work reports results on several continuous control benchmarks, showing better performance among other offline RL algorithms, and a few ablations regarding the action selection method and the ranking function.

**Summary Of The Review:**

Given that the proposed contribution is mostly empirical, the experimental methodology does not provide enough evidence to clearly support the introduced method, as detailed in the weaknesses. Furthermore, the method presents critical limitations and does not seem to satisfy one of its main motivations (policy improvement due to exploration in the online fine-tuning phase). I do not believe that the paper is ready for acceptance, although I am open to changing my review in case of further experiments and some rewriting to better describe the challenges addressed by the proposed work.

---

> ### Author Response · Authors · 2022-11-19
> **Response to Reviewer b1rp**
>
> Thank you for providing valuable feedback on the paper. We will use it to revise the paper. We address the raised issues below:
>
> - Re "The proposed method relies on strong assumptions that are not discussed in the paper": O3F indeed relies on a continuous action space. With respect to a smoothness assumption on the Q-function, our method does not make any additional assumptions. Once a suitable Q-function has been learned for the task, O3F can be applied on top of it to select better actions in expectation.
>
> - Re "It is important to clearly state the challenges mitigated by the proposed method": Online fine-tuning might have two different goals: learning completely new skills that are not in the offline dataset (e.g. handling a completely new type of object); and improving the existing skills that the agent has already learned to improve performance. O3F focuses on the second goal, achieving policy improvement by trying out new behaviors that are proximal to the dataset, instead of trying to discover completely-new behaviors. We will revise Section 5 in the paper to make this more clear.
>
> - Re feedback for Figures 2, 4, and 5: we will include a common response to all reviewers addressing these issues shortly.
>
> - Re feedback for Figure 3: thank you for the suggestion. For the revised version of the paper we will certainly include a comparison with Wu et. al., though as detailed in the general message we will be unable to include this in this rebuttal period.
>
> - Re feedback for Figure 6: Even without using the uncertainty optimism scoring function, the action selection mechanism is still optimistic, because it selects the best estimated action among 100 candidates.
>
> - Re computational cost: thank you for bringing this up. We make use of an efficient implementation of the critic-ensemble using the FuncTorch library, which vectorizes the ensemble operations and makes them essentially as fast as a single model by parallelizing computations. With regards to evaluating 100 actions, notice that this step occurs only when taking a step in the environment (i.e. it’s not 100 actions for each state in a training batch), and 100 actions can be evaluated cheaply using an efficient critic-ensemble implementation.

---

### Author Response · Authors · 2022-11-19
**Common response to all reviewers**

We sincerely thank the reviewers for taking the time to provide valuable feedback on the paper. We are currently working on improving the paper based on the suggestions, particularly the suggestions around the choice of baselines, the mixed results on some environments, and new ablations. But, we unfortunately will not be able to include these additions before the author response deadline.

---

### Decision · Program_Chairs · 2023-01-20

**Decision:**

Reject

**Justification For Why Not Higher Score:**

Reviewers have consensus that the paper should be rejected.

**Justification For Why Not Lower Score:**

N/A

**Metareview: Summary, Strengths And Weaknesses:**

The paper considers the problem of transferring an RL agent trained on offline data to the setting when online data is available.

The reviewers have brought up several issue. Many of them are related to the empirical studies, including not having proper baselines. There was also concerns regarding that the proposed algorithm, O3F, does not address the issues presented as the motivation.

As all reviewers are on the negative side, I cannot recommend acceptance of this paper. I encourage the authors to incorporate reviewers' comments in revising their paper.

From my own reading of the paper, I did not find any information about the number of runs/seeds for each empirical results. Please make sure they are included, and that the number of runs is large enough to have statistically meaningful results.